

# Bacterial communities of the psyllid pest *Bactericera cockerelli* (Hemiptera: Triozidae) Central haplotype of tomato crops cultivated at different locations of Mexico

Maria Goretty Caamal-Chan[1], Aarón Barraza[1], Abraham Loera-Muro[1], Juan J. Montes-Sánchez[2], Thelma Castellanos[3] and Yolanda Rodríguez-Pagaza[4]

[1] Agricultura en Zonas Áridas, CONAHCYT-Centro de Investigaciones Biológicas del Noroeste, SC, La Paz, B.C.S., México
[2] Agricultura, CONAHCYT-Centro de Investigaciones Biológicas del Noroeste, SC, Guerrero Negro, B.C.S., México
[3] Agricultura en Zonas Áridas, Centro de Investigaciones Biológicas del Noroeste, SC, La Paz, B.C.S., México
[4] Universidad Autónoma Agraria Antonio Narro, Saltillo, Coahuila, México

Corresponding authors
Maria Goretty Caamal-Chan, mcaamal@cibnor.mx
Aarón Barraza, abarraza@cibnor.mx

## ABSTRACT

**Background:** The psyllid, *Bactericera cockerelli*, is an insect vector of 'Candidatus Liberibacter' causing "Zebra chip" disease that affects potato and other *Solanaceae* crops worldwide. In the present study, we analyzed the bacterial communities associated with the insect vector *Bactericera cockerelli* central haplotype of tomato crop fields in four regions from Mexico.

**Methods:** PCR was used to amplify the mitochondrial *cytochrome oxidase I gene* (*mtCOI*) and then analyze the single nucleotide polymorphisms (SNP) and phylogenetic analysis for haplotype identification of the isolated *B. cockerelli*. Moreover, we carried out the microbial diversity analysis of several *B. cockerelli* collected from four regions of Mexico through the NGS sequencing of 16S rRNA V3 region. Finally, *Wolbachia* was detected by the *wsp* gene PCR amplification, which is the *B. cockerelli* facultative symbiont. Also we were able to confirm the relationship with several *Wolbachia* strains by phylogenetic analysis.

**Results:** Our results pointed that *B. cockerelli* collected in the four locations from Mexico (Central Mexico: Queretaro, and Northern Mexico: Sinaloa, Coahuila, and Nuevo Leon) were identified, such as the central haplotype. Analyses of the parameters of the composition, relative abundance, and diversity (Shannon index: 1.328 ± 0.472; Simpson index 0.582 ± 0.167), showing a notably relatively few microbial species in *B. cockerelli*. Analyses identified various facultative symbionts, particularly the *Wolbachia* (*Rickettsiales*: *Anaplasmataceae*) with a relative abundance higher. In contrast, the genera of *Sodalis* and 'Candidatus Carsonella' (*Gammaproteobacteria*: *Oceanospirillales*: *Halomonadaceae*) were identified with a relatively low abundance. On the other hand, the relative abundance for the genus 'Candidatus Liberibacter' was higher only for some of the locations analyzed. PCR amplification of a fragment of the gene encoding a surface protein (*wsp*) of

*Wolbachia* and phylogenetic analysis corroborated the presence of this bacterium in the central haplotype. Beta-diversity analysis revealed that the presence of the genus 'Candidatus Liberibacter' influences the microbiota structure of this psyllid species.
**Conclusions:** Our data support that the members with the highest representation in microbial community of *B. cockerelli* central haplotype, comprise their obligate symbiont, *Carsonella*, and facultative symbionts. We also found evidence that among the factors analyzed, the presence of the plant pathogen affects the structure and composition of the bacterial community associated with *B. cockerelli*.

# INTRODUCTION

Arthropods are among the most abundant animals that provide comfortable habitats for microorganisms, harboring a wealth of bacteria that favor their adaptation forming with it a holobiont, which is defined as the relationship of the host and its associated communities of microorganisms (microbiota) (*Simon et al., 2019*). Insects are the dominant group of arthropods, with the orders with the most insect species, Coleoptera, Diptera, Hymenoptera, and Hemiptera (*Stork, 2018*). Like other organisms, insects establish associations with microorganisms, influencing their ecology and evolution (*Weiss & Aksoy, 2011*; *Gurung, Wertheim & Falcao Salles, 2019*).

Several reports on insect microbiota show that plant sap-feeding insects have preserved microbiota communities with high dominance of primary and secondary symbionts, which are acquired by vertical transmission or horizontal acquisition to secure a stable microbiota (*Jing et al., 2014*; *Haag, 2018*; *Cooper et al., 2022*; *Nakabachi, Inoue & Hirose, 2022a, 2022b*). In addition, location, diet, and plant pathogens affect the structure and composition of the insect-associated bacterial community (*Arp et al., 2014*; *Song et al., 2019*; *Moussa et al., 2020*).The ability of phytophagous insects to adapt to nutrient-deficient diets can be attributed to associations with obligate primary symbionts, which provide essential nutrients (amino acids and vitamins) lacking from the host's diet (*Gurung, Wertheim & Falcao Salles, 2019*). Additionally, though secondary facultative endosymbionts generally are not required for host survival, they confer on insect benefits under certain environmental conditions. For example, thermal tolerance, protection against plants' defenses, resistance to insecticides, host immunity, fecundity, and non-essential nutrients supply (*Weeks et al., 2007*; *Teixeira, Ferreira & Ashburner, 2008*; *Osborne et al., 2012*; *Darby et al., 2014*; *De Clerck et al., 2015*; *Chuche et al., 2017*; *Chung et al., 2017*).

The relationship between phytophagous hemipteran psyllids species and their symbionts is one of the most studied. Among the symbionts identified is found, their primary symbiont, "*Candidatus* Carsonella rudii" (*Gammaproteobacteria*). On the other hand, a variety of secondary symbionts have been observed depending on the psyllid species, including *Wolbachia* (*Alphaproteobacteria*:*Rickettsiales*), *Rickettsia*

(*Alphaproteobacteria*: *Rickettsiales*), *Rickettsiella* (*Gammaproteobacteria*: *Diplorickettsiales*), and *Diplorickettsia* (*Gammaproteobacteria*: *Diplorickettsiales*) (*Hosseinzadeh et al., 2019*; *Nakabachi, Inoue & Hirose, 2022a*, *2022b*; *Serbina et al., 2022*; *Martoni et al., 2023*). Insects can act as vectors, that is, the ability of an insect to transmit pathogens. The association between pathogen and vector is a complex network of interactions involving the whole microbial community (*Weiss & Aksoy, 2011*; *Gonella et al., 2019*). Psyllids (Insecta: Hemiptera: Psylloidea) are phloem-feeding. The Liviidae, Psyllidae, and Triozidae families are proposed as pathogenic bacteria's most economically relevant vectors (*Jing et al., 2014*; *Serbina et al., 2022*). The plant pathogens that transmit the psyllids are 'Candidatus Liberibacter spp.' (*Alphaproteobacteria*: *Rhizobiales*) and 'Ca. Phytoplasma spp.' (*Firmicutes*: *Acholeplasmatales*), the reason why they are considered pests (*Cooper et al., 2022*; *Serbina et al., 2022*). The two most devastating pests are *Diaphorina citri* (Psyllidae: Diaphorininae) and *Bactericera cockerelli* (Triozidae), as well as Aphalaridae species (*Munyaneza, 2012*; *Hosseinzadeh et al., 2019*).

The potato or tomato psyllid *Bactericera cockerelli* (Sulc) (Hemiptera: *Triozidae*) is endemic to America, whose population, or the most part is grouped in western North America and other in the central USA and Eastern Mexico (*Butler & Trumble, 2012*). Four different potato psyllid haplotypes or genetic populations have been identified based on nucleotide differences in the *cytochrome oxidase I* (*mtCOI*) gene: the haplotypes are known as western, central, northwestern, and southwestern. The haplotypes are defined in overlapping geographical regions in the United States of America, Mexico, Nicaragua, Salvador, Honduras, and recently in New Zealand and Ecuador. (*Swisher, Munyaneza & Crosslin, 2012*; *Swisher et al., 2013*; *Swisher, Henne & Crosslin, 2014*; *Carrillo, Fu & Burckhardt, 2019*). The psyllid *B. cockerelli* is a severe pest of potato, tomato, and pepper production in Central and North America because it is a vector of 'Ca. Liberibacter solanacearum' is associated with potato zebra chip disease. This disease was first reported in potato fields around Saltillo, Mexico, in 1994, and it was identified later in the United States of America, Honduras, and Nicaragua (*Munyaneza, 2012*; *Swisher et al., 2013*).

Previous studies of potato psyllid revealed that '*Candidatus* Carsonella ruddii', *Acinetobacter* sp., *Wolbachia* sp., and 'Ca. Liberibacter solanacearum' as part microbial communities of *B. cockerelli* (*Nachappa et al., 2011*; *Hail, Dowd & Bextine, 2012*; *Arp et al., 2014*; *Cooper et al., 2022*). Researches in insect microbial ecology has shown that in some psyllids species, the host biology and geography lead to changes in the bacterial community structure (*Serbina et al., 2022*). Crop variety and collection year do not affect the microbial community of *B. cockerelli*. However, geographical location and haplotypes if they have an (*Arp et al., 2014*). The highest degree of similarity in bacterial communities was observed between the Central and Western haplotypes of *B. cockerelli* despite being collected in different geographic regions (*Cooper et al., 2022*). In recent decades, there has been increasing evidence that the microbiome may also influence the capacity of vector to transmit pathogens (*Weiss & Aksoy, 2011*; *Fagen et al., 2012*; *Eleftherianos et al., 2013*; *Jiang et al., 2023*). Various studies showed that in the Asian citrus psyllid, *Diaphorina citri* Kuwayama, vector for 'Ca. Liberibacter asiaticus', the presence of the plant pathogenic bacteria was a significant external factor that could influence the psyllid bacterial

community (*Fagen et al., 2012*; *Jiang et al., 2023*). This adds another element that adds to the importance of describing the bacterial communities of insect vectors. For instance, haplotypes of psyllids differ in the level of *Wolbachia* carriage; northwestern and southwestern haplotypes are uninfected of *Wolbachia*, but western and central haplotypes carry strains A and B of *Wolbachia* (*Cooper et al., 2015*). In recent decades, there has been increasing evidence that the microbiome may also influence the capacity of vector to transmit pathogens (*Weiss & Aksoy, 2011*; *Fagen et al., 2012*; *Eleftherianos et al., 2013*; *Jiang et al., 2023*).

In northern and central Mexican regions with intensive agriculture, the Potato zebra chip disease has persisted since its first report (*Rojas-Martínez et al., 2016*). Meanwhile, the potato psyllid, *B. cockerelli*, is present in various parts of Mexico, and little is known about the microbial community associated in these locations. Therefore, this work examined the bacterial profiles within the *B. cockerelli* psyllid samples from northern and central Mexican to identify the microbiota that they might carry.

# MATERIALS AND METHODS

## *Bactericera cockerelli* collecting

We collected adult psyllids of *B. cockerelli*, sexing was not performed for this study. Psyllids were collected from potato (*Solanum lycopersicum*) crops under greenhouse conditions and open field cultivation in four Mexican localities between 2015–2017 (Table 1). The locations of the crops were Nuevo León (24°49′04.9″ N, 100°04′32.0″ W), Sinaloa (25°27′54.4″ N, 108°26′11.6″ W), and Coahuila (25°29′00.1″ N, 100°56′52.2″ W) in northern Mexico and Querétaro (20°34′21.7″ N, 100°25′34.2″ W) in central Mexico (Fig. 1). Psyllids in the adult stage were identified for the ground colour white yellowing in the thorax with a large well defined brown marking (*Sulc, 1909*; *Munyaneza, 2012*). The psyllid collecting was carried out during their feeding periods, placed in 95% (v/v) ethanol and stored at −20 °C until total DNA extraction.

## DNA extraction

Fourteen psyllids were collected obtained two pools per localization (a total of eight samples * 7 insects × pool = 14 insects/location) (Table 1). We washed each insect pool three times with 70% (v/v) ethanol, decanting the supernatant to remove environmental contaminants before DNA extraction. After, the samples were processed as described by *Caamal et al. (2019)*. The quality of the extracted DNA was assessed using a NanoDrop 2000c spectrophotometer (Thermo Fisher Scientific, Waltham, MA, USA). Finally, the extracted DNA was stored at −20 °C for further analysis.

## Detection and identification of *B. cockerelli* haplotypes (*mtCOI* gene)

The *mtCOI* gene from psyllid-DNA was amplified by PCR, using primers pairs CO1F3/CO1R3 reported by *Crosslin, Lin & Munyaneza (2011)*. PCR reactions were performed using the manufacturer's instructions for Go Taq DNA Polymerase (Promega, Madison, WI, USA) and was used as temperate 100 ng of DNA. The thermocycling profile included an initial denaturation step for 5 min at 95 °C followed by amplification conditions

**Table 1 Descriptive information of each sample used in this study.**

| Sample ID | Host stage | Location | Crop conditions | Host plant | Collection date day-month-year |
|---|---|---|---|---|---|
| BcAdGHNL01 | Adult | Nuevo León | Greenhouse | *S. lycopersicum* | 21-02-2015 |
| BcAdGHNL02 | Adult | Nuevo León | Greenhouse | *S. lycopersicum* | 22-02-2015 |
| BcAdOFSin01 | Adult | Sinaloa | Open field | *S. lycopersicum* | 21-05-2017 |
| BcAdOFSin02 | Adult | Sinaloa | Open field | *S. lycopersicum* | 22-05-2017 |
| BcAdGHQue01 | Adult | Querétaro | Greenhouse | *S. lycopersicum* | 02-10-2017 |
| BcAdGHQue02 | Adult | Querétaro | Greenhouse | *S. lycopersicum* | 02-10-2017 |
| BcAdOFCoa01 | Adult | Coahuila | Open field | *S. lycopersicum* | 06-10-2016 |
| BcAdOFCoa02 | Adult | Coahuila | Open field | *S. lycopersicum* | 06-10-2016 |

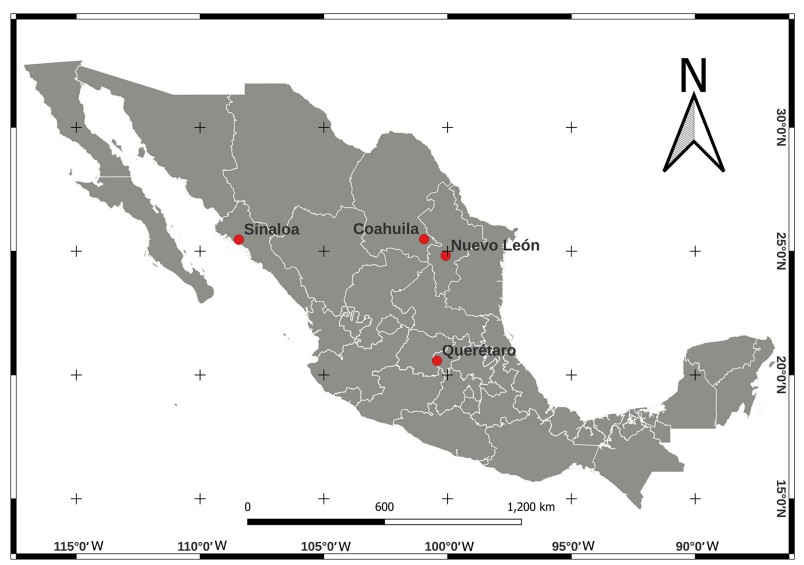

**Figure 1 Location map for *Bactericera cockerelli* sampling in Northern Mexico (Nuevo Leon, Coahuila, and Sinaloa) and in Central Mexico (Queretaro) crops of tomato.**

for 35 cycles (95 °C, 30 s; 55 °C, 1 min, and 72 °C, 1 min), and a extension condition at 72 °C for 5 min. PCR product was purified using Zymoclean™ Gel DNA Recovery Kit (Irvine, CA, USA), and Sanger sequenced in triplicate. The partial COI sequences obtained of Nuevo León, Sinaloa, Coahuila, and Querétaro locations were employed to identify the haplotype based on the analysis of a single nucleotide polymorphism (SNP) that existed in the *mtCOI* sequences between psyllids, as was described in *Swisher, Munyaneza & Crosslin (2012)*. Previously identified sequences were used as reference: Western haplotype (JQ708095, AY971885) and Central haplotype (JQ708094, FJ175374, EF372597). Phylogenetic analyses of the sequences *mtCOI* gene obtained from this work were performed. The *mtCOI* genes sequences of psyllid used in this study were: Northwestern: JQ708093.1, KX510008.1, and KR534768.1; Central and Western: KX130767.1, JQ708094.1, JQ708095.1, FJ175374.1, EF372597.1, MK054304.1, Southwestern: KC305359.1.

We used the gene sequence KP223855.1 of *Bactericera maculipennis* as an outgroup sequence. The phylogenetic analysis was performed with MUSCLE alignments with default parameters followed by phylogenetic tree were inferred by the maximum likelihood methodology coupled with Kimura-2 parameter model, with a bootstrapping of 1,000 replicates for statistical support of lengths of evolutionary branches, employing the Molecular Evolutionary Genetics Analysis software package version 7 (MEGA7) (*Kumar, Stecher & Tamura, 2016*).

## Detection of *Wolbachia* and phylogenetic analysis of endosymbiont *Wolbachia*

The detection of the endosymbiont *Wolbachia* was carried out by PCR using the primer sequences for the gene encoding a surface protein of *Wolbachia* (*wsp* 81F/*wsp* 691R) reported in *Zhou, Rousset & O'Neil (1998)*. PCR amplification was performed with the same reagents and concentrations described in the previous section, with 100 ng of DNA. The PCR program was used under the thermocycling conditions described in the previous section. The PCR products were purified using Zymoclean™ Gel DNA Recovery Kit (Irvine, CA, USA), and Sanger sequenced in triplicate. Phylogenetic analysis of *wsp* was performed as described previously. The *wsp* gene sequences of *Wolbachia* used here were: Strain A: EU916190.1, GU014541.1, AB094372.1, JQ837254.1, KM267306.1, and AY971950.1; Strain B AY971930.1, AY971915.1, KM267307.1, AY971925.1, AY971917.1, AY971927.1, MK550508.1, MK534180.1, KP208732.1, GQ385975.1, KP208725.1, AB094362.1, AB094355.1, KP208729.1; and Strain c AJ252176.1, AJ252178.1, AJ252062.1; and Strain d: AJ252177.1, AJ252061.1, AJ252175.1 sequences of *Wolbachia* of Nematoda phylum.

## 16S V3 rDNA sequencing

Seven psyllids was pooled (DNA pools). A total of eight DNA pools (2 pools per location) were used to amplify the hypervariable region of the bacterial 16S rRNA. The V3 region was amplified by PCR (V3-338F and V3-533R) (*Huse et al., 2008*) the primers contain Illumina adapters. Sequences were obtained on the Illumina MiniSeq platform in a $2 \times 150$ bp (300 cycles).

## Microbial diversity analysis

We applied the modified algorithm of Mott for the sequencing reads, and the threshold phred (Q) value was established at 33 (*Ewing et al., 1998*; *Ewing & Green, 1998*). Then the resulting sequences were paired-end, and we used BBmerge to merge them (*Kearse et al., 2012*; *Bushnell, Rood & Singer, 2017*). We used the Geneious assembler for *de novo* assembling of the merged sequences with an identity cut-off at 98% (Geneious Prime 2019.2.3; www.geneious.com). We applied Megablast for contigs, unassembled reads, and with the GenBank database non-redundant ("*nt-nr*") for each pooled sample (*Randle-Boggis et al., 2016*). These results served as the foundation to establish a specific database on a pooled sample basis. The Sequence Classifier applied a 99% identity threshold for the taxonomical assignation of the sequence reads per pooled sample. It uses its specific
database to yield the frequencies into the operational taxonomic units (OTUs) table (Geneious Prime 2019.2.3; www.geneious.com). We used the R programming language (*R Core Team, 2022*), starting with the use of the best hits frequencies for each OTU divided by the hits total amount for each pooled sample for data normalization and applying several packages (vegan, iNEXT, pheatmap) and tailored scripts for ecological data analyses, such as alpha indices (Chao1, Shannon, and Simpson), Welch t-test was performed to compare between the 'Ca. Liberibacter' relative abundance (high- and low-carrier), Bray-Curtis dissimilarity distance estimations, and principal coordinates analysis were performed for beta diversity analyzes. The PERMANOVA statistical analysis was performed to determine the significance of microbial community differences. And, the differential abundance OTUs (DAOTUs) analysis of the pooled samples (*Oksanen et al., 2014*; *Hsieh, Ma & Chao, 2016*).

## RESULTS

### Sequence analysis of *mtCOI* gene for identification of haplotype of *B. cockerelli*

All psyllids' samples were assessed through PCR using the primers specific to the *B. cockerelli mtCOI* gene. The resulting PCR products for all samples were positive with an amplicon length of approximately 455 bp (Fig. 2A). The *mtCOI* amplicon was sequenced and subjected to analysis to identify the haplotype of the psyllids in search of only one single-nucleotide polymorphism (SNP) at nucleotide 51 that exists between the Western (nucleotide 51:C) and Central (nucleotide 51:T) haplotypes in a 455-bp fragment of the *mtCOI* gene (Fig. S1). With the consensus sequence length of 453 nucleotides, the *mtCOI* sequences obtained in this work align from the nucleotide 48 up to nucleotide 500 with the reference *mtCOI* sequences numbers: Western (JQ708095 and AY971885), and Central (JQ708094, FJ175374, and EF372597). All the samples of this study had thymine in the residue corresponding to the nucleotide 51 of the reference sequences represented the Central haplotype of the psyllid (Fig. S1).

In the maximum likelihood (ML) tree, *mtCOI* sequences identified in the present study formed a robustly supported clade (bootstrap: 94%) with those of psyllids belonging to the Central and Western haplotype, with a close relationship with the Central haplotype (JQ708094.1 and MK054304.1), consistent with the results of the analysis of SNP (Fig. 2B).

### Detection and identification of symbionts *Wolbachia* from *B. cockerelli* Central haplotype

The *Alphaproteobacteria Wolbachia* was detected using PCR that amplifies a fragment of the gene encoding a surface protein of *Wolbachia* (*wsp*) in all *B. cockerelli* samples collected in this work (Fig. 3A). The PCR products were sequenced, and the *wsp* sequences were analyzed with blastn algorithm for global alignment, with 100% identity to *Wolbachia* endosymbiont of *B. cockerelli* (GenBank accession number: KM267307; *Cooper et al., 2015*) (Fig. 3A). The phylogenetic analysis clustered the *Wolbachia* of *B. cockerelli* (Central haplotype) from Querétaro, Coahuila, Nuevo León, and Sinaloa with the monophyletic

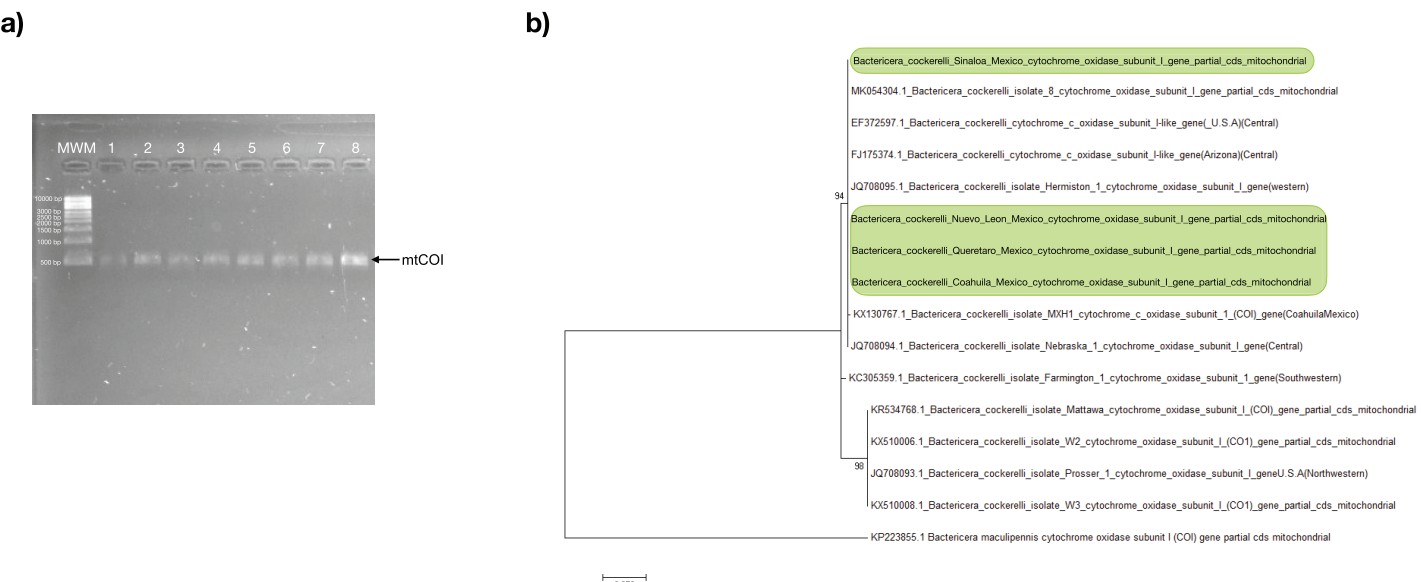

**Figure 2** *Bactericera cockerelli* **Central haplotype identification.** (A) Amplification products of *mtCOI* of *Bactericera cockerelli* (≈ 0.5 kb); lane 1-2. BcAdGHNL01-02; lane 3-4. BcAdOFSin01-02; Lane 5-6. BcAdGHQue01-02; Lane 7-8. BcAdOFCoa01-02. MWM. Molecular weight marker for DNA (Direct Load 1 kb DNA Ladder Sigma); (B) Phylogenetic relationships inferred from maximum likelihood based on *mtCOI* fragment of *Bactericera cockerelli* collected in different locations from Mexico, and selected sequences from GenBank. The number on the branches represent bootstrap values for 1,000 replicates. *Bactericella maliculipenni*s were used as outgroup sequence. Sequences colored in green correspond to this study.

grouping of *Wolbachia* strain B from *B. cockerelli* (GenBank accession number: KM267307; *Cooper et al., 2015*) (Fig. 3B). In addition, the sequences from the samples analyzed here were related to the monophyletic grouping *Wolbachia* strain B from other insects (*e.g.*, GenBank accession number: GQ385975.1, AB094355, and AB094362.1), and the paraphyletic group of *Wolbachia* strain A identified in *B. cockerelli* (GenBank accession number: KM267306; *Cooper et al., 2015*) (Fig. 3B).

## Sequencing run metrics and microbial community diversity analyzes from *B. cockerelli*

We obtained 830,240 reads of eight samples of adult psyllids collected in Querétaro, Coahuila, Nuevo León, and Sinaloa. After bioinformatic processing (trimming, quality control, and paired-end merging) the whole sequences we kept was 399,936 reads that, in turn, were classified into 689 OTUs at 99% sequence identity (Table 2). It is worth noting that *B. cockerelli* samples might carry bacterial plant pathogens. Among the identified OTUs, we found the presence of the bacterial plant pathogen 'Candidatus Liberibacter'. Moreover, the OTU corresponding to 'Candidatus Liberibacter' is among the most abundant OTUs. We performed an unsupervised hierarchical biclustering analysis of the most abundant OTUs (>1% relative abundance/sample). This analysis showed the grouping of the samples mainly based on the relative abundance of eight OTUs (high and low relative abundance), among which 'Candidatus Liberibacter' OTU is present (Fig. 4). Then, we assigned the samples into two well-defined groups, high- (BcAdOFCoa01, BcAdOFCoa02, BcAdGHNL02, and BcAdGHQue01) and low-carriers (BcAdGHNL01,

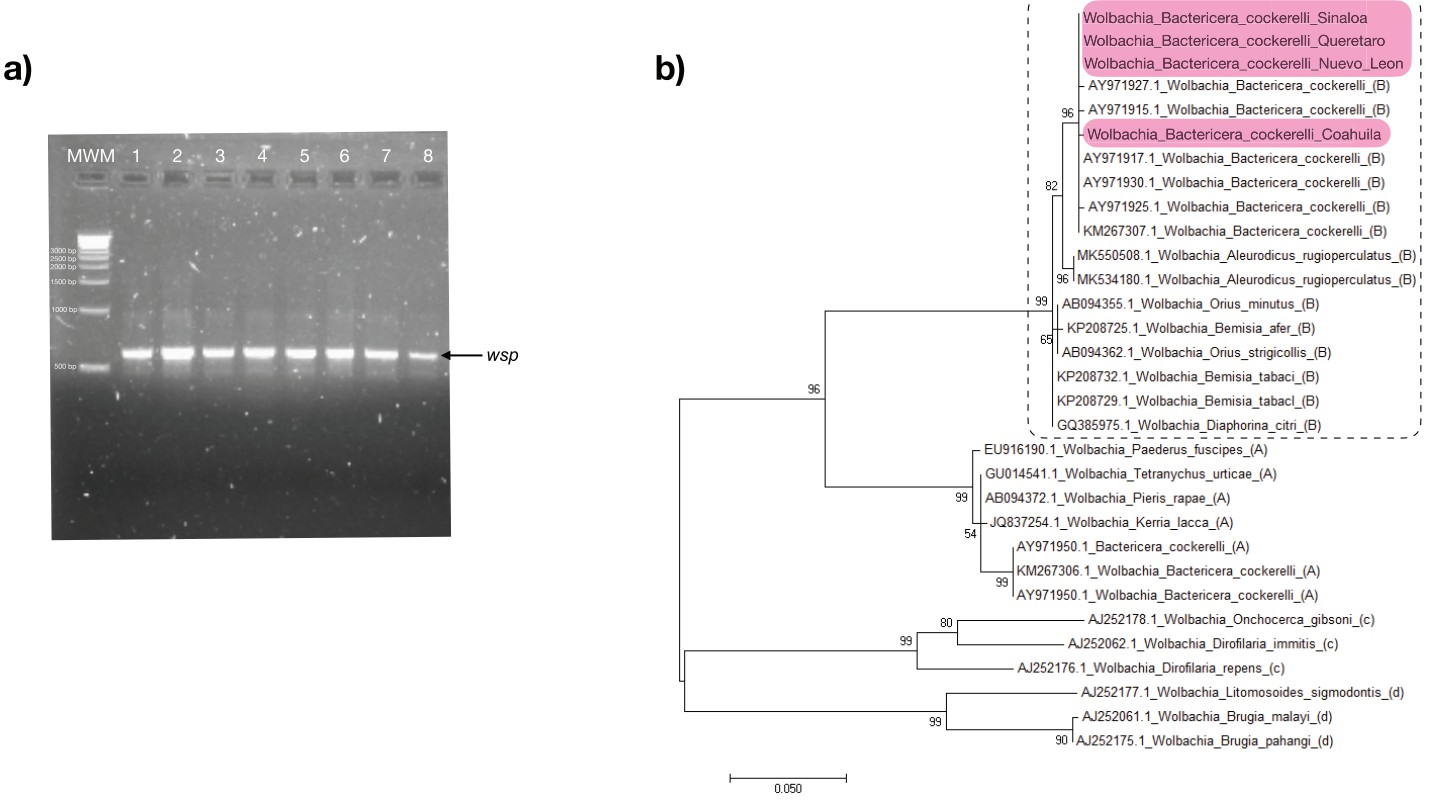

**Figure 3 *Wolbachia* strain B identification.** (A) *wsp* amplification of collected *B. cockerelli* (≈ 0.6 kb); lane 1-2. BcAdGHNL01-02; lane 3-4. BcAdOFSin01-02; Lane 5-6. BcAdGHQue01-02; Lane 7-8. BcAdOFCoa01-02. MWM. molecular weight marker for DNA (Direct Load 1 kb DNA Ladder Sigma). (B) Phylogenetic relationships inferred from maximum likelihood based on sequences *wsp* of *Wolbachia* strains identified in insects, selected reference sequences from GenBank. The number on the branches represent bootstrap values for 1,000 replicates. Sequences colored in pink correspond to this study. Dashed lines represent subgroup B.

BcAdGHQue02, BcAdOFSin01, and BcAdOFSin02), based upon the relative abundance of 'Candidatus Liberibacter'. Moreover, we assessed the alpha diversity indices through inter- and extrapolation sampling curves analysis. The estimation of diversity (Chao1 index, $q = 0$), richness (Shannon index, $q = 1$), and evenness (Simpson index, $q = 2$) showed differences among insect samples also due to 'Candidatus Liberibacter' relative abundance (high- and low-carrier) independently of the location (Fig. 5). Also, the overall alpha diversity indices were estimated from the bacterial community abundance to assess diversity (observed OTUs and Chao1 index), richness (Shannon index), and evenness (Simpson index) for the *B. cockerelli* Central haplotype samples. For all samples, the observed OTUs and Chao1 index, Shannon index, and Simpson index indicated a high specialization of the bacterial community associated with *B. cockerelli* (85 ± 16.80, 165.96 ± 29.935, 1.328 ± 0.472, and 0.582 ± 0.167, respectively) (Table 2 and Fig. S2). It was also possible to infer a grouping based on 'Candidatus Liberibacter' relative abundance since the Shannon index values above 1.417 to 1.951 and the Simpson index values above 0.6362 to 0.7996 corresponding to the samples with high relative abundance of 'Candidatus Liberibacter' (Table 2 and Fig. S2).

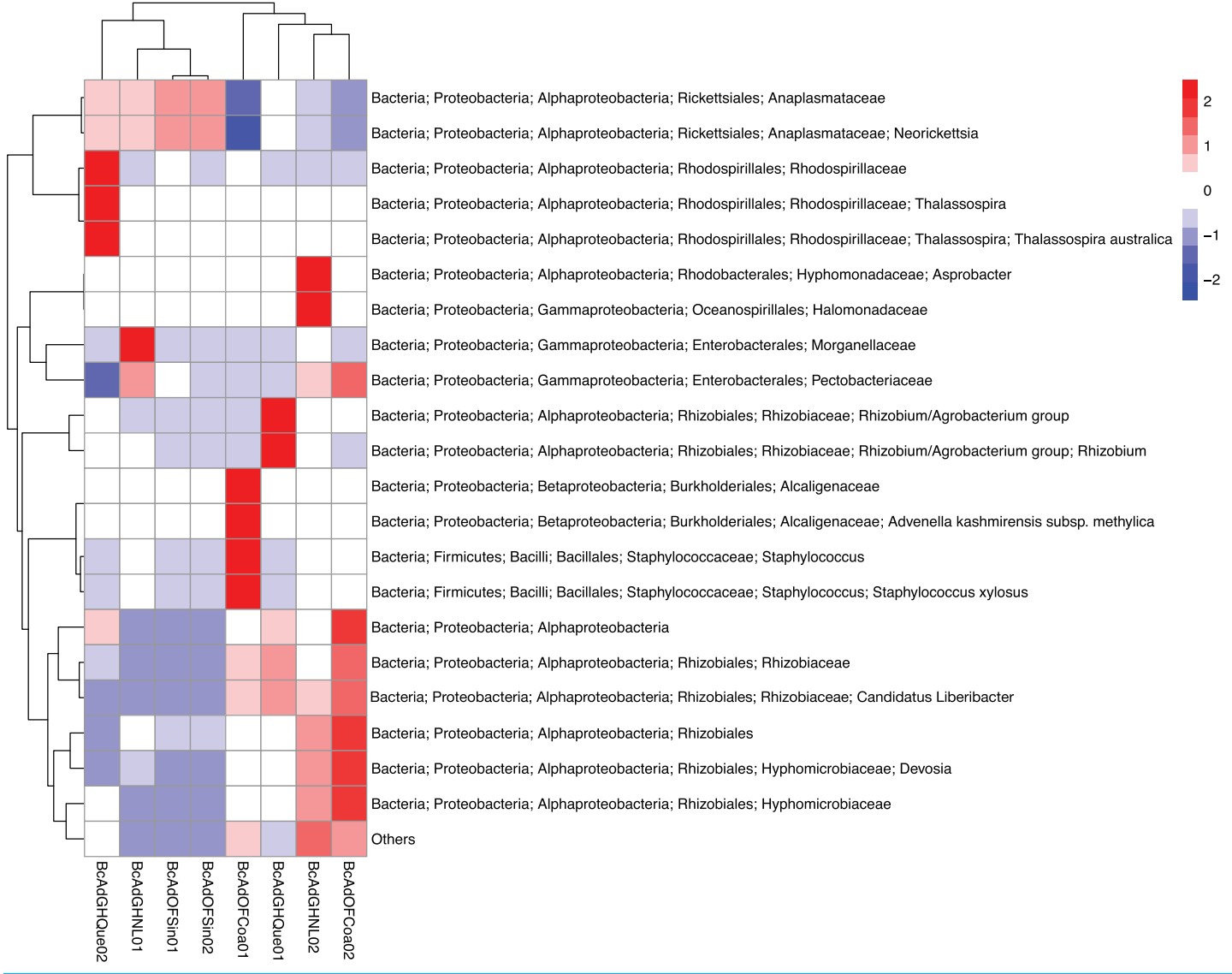

**Figure 4 Heatmap based on the unsupervised hierarchical biclustering of the psyllid samples.** The OTUs with the highest relative abundance were used to perform the analysis in which was able to determine the samples' microbial communities underwent a clustering in a relative abundance-dependent behavior.

We proceeded with the sample clustering analysis or β-diversity analysis applying a Principal Coordinate analysis (PCoA) based on a Bray-Curtis dissimilarity distances estimations between samples. The PCoA analysis showed two discrete groups based on the relative abundance of 'Candidatus Liberibacter' (high- and low-carriers) from *B. cockerelli* Central haplotype samples (Fig. 6). Furthermore, we carried out the Permutational multivariate analysis of variance (PERMANOVA) to determine whether the relative abundance of 'Candidatus Liberibacter' (high- and low-carriers) or location and even crop conditions (open field or greenhouse) played a significant role in the clustering of the samples. The PERMANOVA results showed that 'Candidatus Liberibacter' relative abundance (high- and low-carriers) was the main factor influencing the bacterial

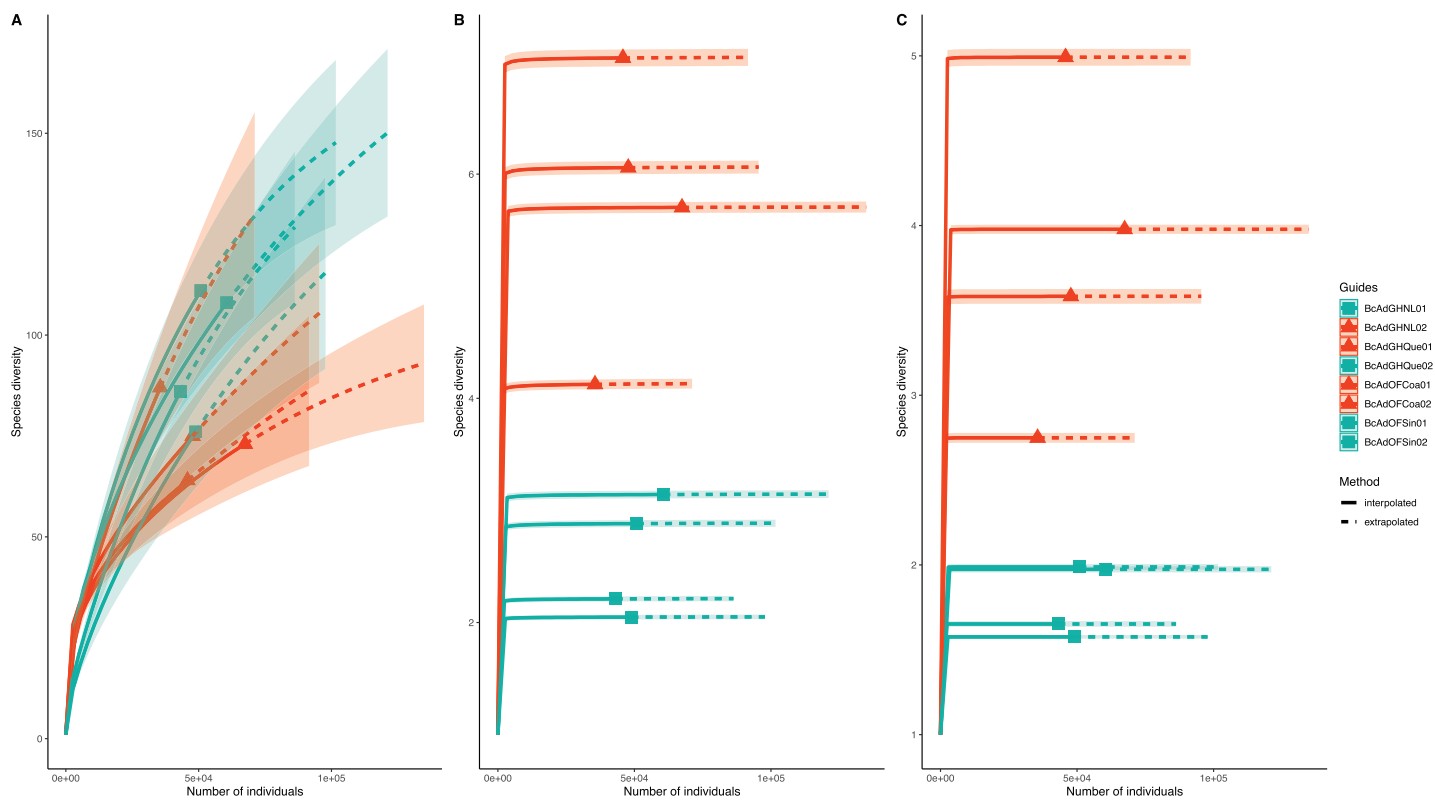

**Figure 5 Inter- and extrapolation (R/E) sampling curves analysis for all insect samples microbial communities.** (A) Estimation of Chao1 index (order $q = 0$), (B) Shannon index (order $q = 1$), and (C) Simpson index (order $q = 2$). Colour code: orange for samples with a high relative abundance of '*Ca.* Liberibacter', seagreen for samples with a low relative abundance of '*Ca.* Liberibacter'.

**Table 2 Sequencing analysis of V3 region of 16S rRNA gene and alpha diversity indices calculated for microbial communities associated with *Bactericera cockerelli*.**

| Sample ID | No. of reads | Location | Crop conditions | Observed OTUs | Chao1 | Shannon | Simpson | '*Ca.* Liberibacter' relative abundance |
|---|---|---|---|---|---|---|---|---|
| BcAdGHNL01 | 121,120 | Nuevo Leon | Green house | 111 | 170.40 | 1.058 | 0.4969 | Low |
| BcAdGHNL02 | 111,358 | Nuevo Leon | Green house | 75 | 174.17 | 1.801 | 0.7208 | High |
| BcAdOFSin01 | 83,190 | Sinaloa | Open field | 86 | 170.18 | 0.793 | 0.3948 | Low |
| BcAdOFSin02 | 139,086 | Sinaloa | Open field | 76 | 174.00 | 0.717 | 0.3655 | Low |
| BcAdGHQue01 | 106,388 | Queretaro | Green house | 87 | 189.21 | 1.417 | 0.6362 | High |
| BcAdGHQue02 | 155,964 | Queretaro | Green house | 108 | 203.40 | 1.145 | 0.4935 | Low |
| BcAdOFCoa01 | 99,858 | Coahuila | Open field | 64 | 139.00 | 1.951 | 0.7996 | High |
| BcAdOFCoa02 | 113,276 | Coahuila | Open field | 73 | 107.36 | 1.741 | 0.7486 | High |
| Mean ± SEM | 116,280 ± 22,722.2 | – | – | 85 ± 16.80 | 165.96 ± 29.935 | 1.328 ± 0.472 | 0.582 ± 0.167 | – |

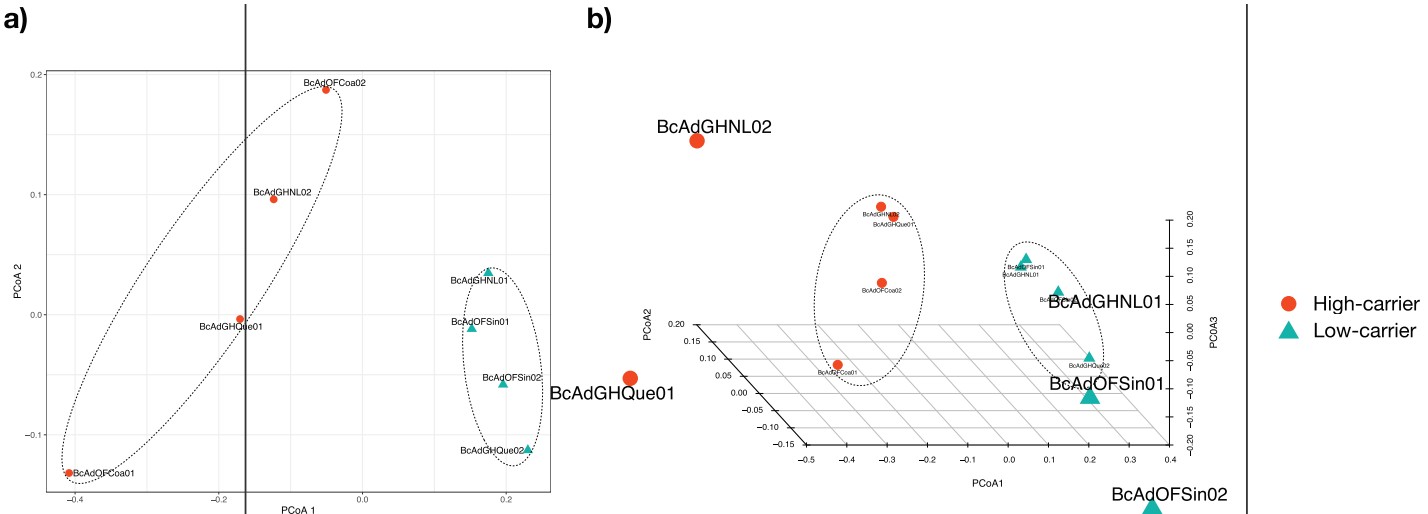

**Figure 6 Principal Coordinate analysis (PCoA) for all insect samples from several locations in Mexico.** The insect samples' microbial communities were clustered in two discrete groups according to high- and low-carrier of the relative abundance of '*Ca*. Liberibacter'. (A) Two coordinates (PCo1 and PCo2), (B) Three coordinates (PCo1, PCo2, and PCo3).

communities' assemblages of *B. cockerelli* central haplotypes samples ($R^2 = 0.58536$, $P = 0.025$), and location and crop condition (greenhouse or open field) did not have any significant effect ($R^2 = 0.45264$, $P = 0.39$; and $R^2 = 0.03368$, $P = 1$, respectively). Furthermore, we extended the PERMANOVA analysis to determine the interactions between '*Candidatus* Liberibacter' relative abundance and location and '*Candidatus* Liberibacter' relative abundance and crop condition. Both PERMANOVA interactions analyzes showed that '*Candidatus* Liberibacter' relative abundance (relative abundance: location, $R^2 = 0.58536$, $P = 0.048$; relative abundance:crop condition, $R^2 = 0.58536$, $P = 0.019$) was the main factor that affected the *B. cockerelli* microbial structure assemblage above location and crop condition ($R^2 = 0.15666$, $P = 0.844$; and $R^2 = 0.03680$, $P = 0.665$, respectively).

The microbial community structures of *B. cockerelli* central haplotype across all samples showed that the dominant phylum was *Proteobacteria* with relative abundances ranging from 91 to 99%. At Class level the dominant class was *Alphaproteobacteria* with a relative abundance ranging from 95 to 99%. At the genus level the dominance of the *Proteobacteria* phylum followed the same trend (relative abundance from 87 to 99%), and the most abundant genera were *Neorickettsia* and '*Candidatus* Liberibacter' ranging from 18% to 98%, and from 0.01% to 66%. The '*Candidatus* Liberibacter' high-carrier samples (BcAdOFCoa01, BcAdOFCoa02, BcAdGHNL02, and BcAdGHQue01) showed relative abundances ranging from 51% to 66%, giving rise to be the most abundant genus in those samples, and the low-carrier samples (BcAdGHNL01, BcAdGHQue02, BcAdOFSin01, and BcAdOFSin02) showed relative abundances ranging from 0.01% to 2.26%, in these samples *Neorickettsia* genus was dominant with relative abundances ranging from 64 to 98% (Fig. 7A). To determine which OTUs underwent significant relative abundance changes in the microbial assemblage across all samples, we carried out a differential abundance OTUs (DAOTUs) analysis. Surprisingly, from the 689 OTUs assessed through the DAOTUs

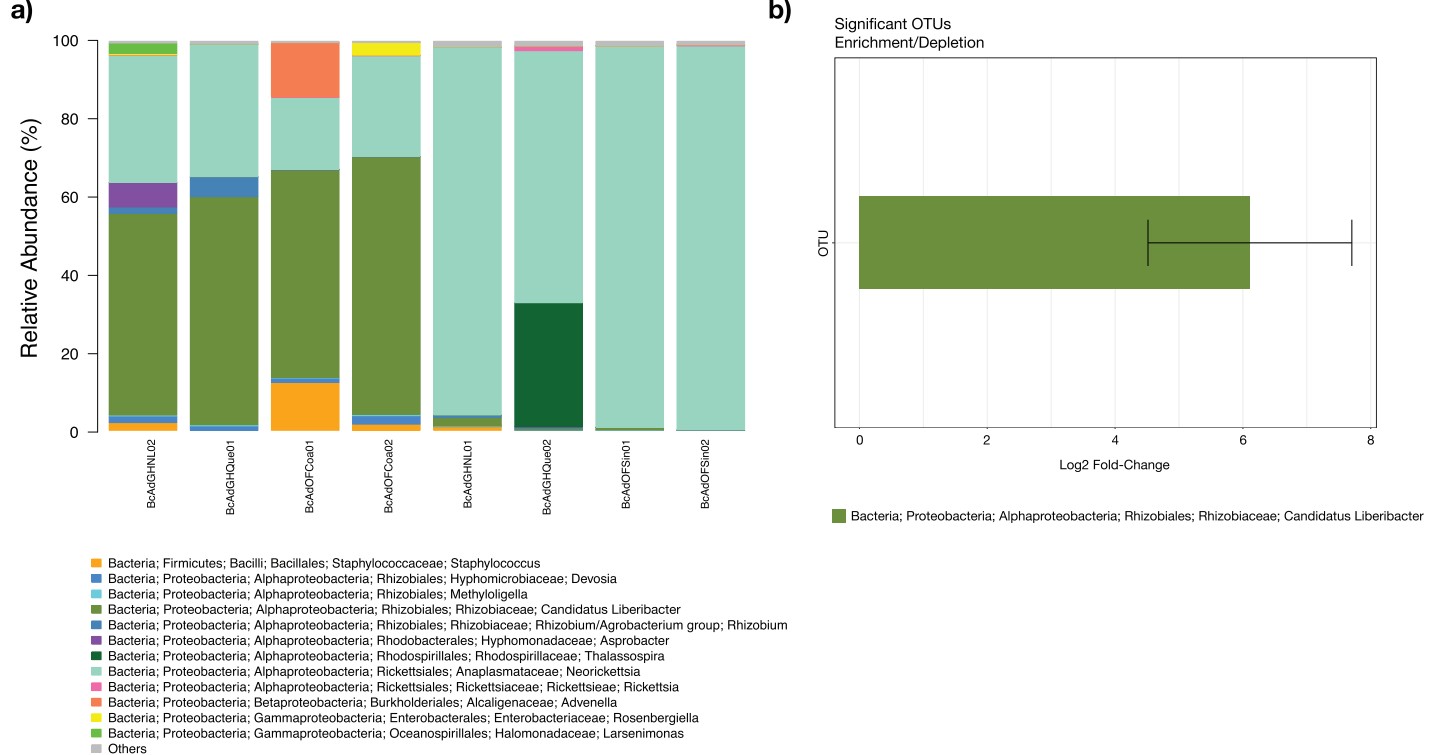

**Figure 7 Microbial structure assemblage of *Bactericera cockerelli* and DAOTUs analysis.** (A) Microbial structures for all psyllid samples at genus level. OTUs with relative abundances <1% are summed as "others". (B) DAOTUs analysis, the barplot is showing the mean log2 fold-change and the error bar is showing ± SD of the differential abundance OTUs.

analysis, only the OTU corresponding to ´Candidatus Liberibacter´ showed significant differential abundance between high- and low-carrier samples (log2foldchange = 6.10, *P value* = 0.000127, *Padj* = 0.0409) (Fig. 7B). Altogether, the presence and the relative abundance of 'Candidatus Liberibacter' suggest a pivotal role in microbial assemblage modelling.

## DISCUSSION

In recent years, there has been an upsurge in the investigations on the microbiome of phytophagous insects, including psyllids that can act as vectors of plant pathogenic bacteria (*Nakabichi, Inoue & Hirose, 2022a*, *2022b*; *Martoni et al., 2023*). The psyllid *B. cockerelli* is one of the pests with a significant economic impact on Solanaceae family production (*i.e.*, *Capsicum annuum*, *Solanum lycopersicon*, and *S. tuberosum*) in North America, including Mexico (*Rojas-Martínez et al., 2016*; *Reyes-Corral et al., 2021*). In the current study, we identified the regional potato psyllid haplotypes. Also, we assessed the microbial communities associated with adult psyllids collected from tomato agricultural fields in different geographical locations of Mexico, with particular attention to the pathogen 'Candidatus Liberibacter' and the symbiont *Wolbachia*.

The distribution in Mexico of at least two of the four haplotypes of *B. cockerelli* has been reported previously: the western haplotype in Baja California peninsula Mexico, and the central haplotype in central and east Mexico (*Liu, Trumbre & Stouthamer, 2006*;
*Swisher et al., 2013*; *Swisher et al., 2018*). Central and western haplotypes are significantly different in developmental traits depending on the host plant (*Mustafa et al., 2015*); however, they are biologically similar in *Wolbachia* infection (*Cooper et al., 2015*). In this study, all psyllids collected in central (Queretaro) and northern Mexico (Sinaloa, Coahuila, and Nuevo Leon) were identified as the central haplotype (Fig. 2B). These results are consistent with several studies have shown that in wild populations of *B. cockerelli* north and central from Mexico the central haplotype were predominate. For example, the haplotype of collected psyllids from an experimental tomatillo plot in a region of northern Mexico (Saltillo, Coahuila) was the central haplotype that predominately occurs (*Reyes-Corral et al., 2021*). Our results of analyzing the fragment of the gene encoding a surface protein of *Wolbachia* (*wsp*) (Fig. 3A) corroborated the presence of this bacterium in the psyllids, one of the biological characteristics of the Central haplotype.

*Wolbachia* (*Rickettsiales*: *Anaplasmataceae*) is among the arthropods' most widely distributed endosymbionts grouped in the *Alphaproteobacteria* bacterial class. *Wolbachia* is currently classified into supergroups A–Q. The supergroups which most commonly infect arthropods are A and B (*Kaur et al., 2021*). Previously, in a specific study with four haplotypes of B. cockerelli, using the primer set for the *wsp* gene was identified to the A and B strains of Wolbachia infecting both western and central haplotypes (*Cooper et al., 2015*). The phylogenetic relationship with other recognized strains using the nucleotide sequence of the *wsp* gene was examined during this study. Results showed a high identity with strain B (Fig. 3B), which is consistent with what is observed in previous reports in which all *Wolbachia* strains previously detected in 12 psyllids species of the family Psyllidae belonged to supergroup B (*Nakabachi, Inoue & Hirose, 2022a*, *2022b*; *Serbina et al., 2022*). A recent study of two psyllids species of the family Triozidae native to North America, including *B. cockerelli*, presented the same results (*Cooper et al., 2022*).

Notably, simple bacterial communities have been reported for sap-feeding insects that present specific bacterial associations reflected in their richness indices (*i.e.*, psyllids, aphids, and whiteflies) (*Jing et al., 2014*; *Hosseinzadeh et al., 2019*; *Morrow et al., 2020*; *Nakabichi, Inoue & Hirose, 2022a*, *2022b*; *Serbina et al., 2022*) concerning other insects with different feeding behaviors, such as Drosophila (*Poff et al., 2017*). Starting the parameters of alpha-diversity, adults of *B. cockerelli* (family Triozidae) in this study showed a notably relatively few microbial species (Table 2). 'Ca. Carsonella' (*Gammaproteobacteria*: *Oceanospirillales*: *Halomonadaceae*), *Sodalis*, *Wolbachia* (*Rickettsiales*: *Anaplasmataceae*) and 'Ca. Liberibacter solanacearum' (*Alphaproteobacteria*) were the most abundant bacterial genera found in our analysis (Fig. 7A). '*Ca*. Carsonella' plays an essential role in synthesizing amino acids lacking from the host´s diet relationship of obligate symbioses (*Nakabachi et al., 2006*). *Wolbachia* plays critical roles on host immunity (*Teixeira, Ferreira & Ashburner, 2008*; *Osborne et al., 2012*), immunocompetence (*Braquart-Varnier et al., 2008*), fecundity (*Weeks et al., 2007*), and metabolic activity (*Darby et al., 2014*). Notably simple bacterial communities like these have been reported for other psyllids species, including the different haplotypes of *B. cockerelli* (*Morrow et al., 2020*; *Štarhová Serbina et al., 2022*; *Cooper et al., 2022*; *Liu et al., 2022*; *Nakabichi, Inoue & Hirose, 2022a*, *2022b*; *Jiang et al., 2023*).

The analysis of the abundance of the different taxa in the psyllids revealed microbial communities with low diversity, in which only some genera were present with high abundance. In particular, the samples with the most abundant OTUs of 'Ca. Liberibacter' present a value higher in the Shannon index (SF2). These findings are in accordance with previous reports where populations of Asian citrus psyllid (*Diaphorina citri*) are analyzed infected and non-infected with 'Ca. Liberibacter asiaticus' infection significantly increased in higher alpha diversity metrics than in non-infected individuals (*Liu et al., 2022*; *Jiang et al., 2023*). Similar results were observed in insects of the Cicadellidae and Cixiidae family (*Euscelis incisus*, *Euscelidius variegatus*, and *Hyalesthes obsoletus*) infected with 'Candidatus Phytoplasma solani' (*Moussa et al., 2020*). As in those studies, it also was possible to define two discrete groups based on the 'Ca. Liberibacter' relative abundance (Figs. 5–7) supported the unsupervised hierarchical clustering, Bray-Curtis distance estimations, and PERMANOVA analyzes. The results suggest that the presence of the pathogen affect the composition of the community in these psyllids.

The presence of plant pathogenic bacteria in the vector insects affects the abundance of some bacterial genera (*Fagen et al., 2012*; *Hosseinzadeh et al., 2019*; *Moussa et al., 2020*; *Liu et al., 2022*; *Jiang et al., 2023*). In the present work, we determined that the samples with more presence of the 'Ca. Liberibacter' (high-carriers) had a lower abundance in *Wolbachia*. These results are by reports that have noted that in insect vectors of 'Ca. Phytoplasma solani' the infection for *Phytoplasma* may have adverse effects on the endosymbionts *Sulcia* and *Wolbachia* in terms of abundance. In the said study, *Wolbachia* was reduced in the infected insect species, except in *Dictyophara europaea*, where the abundance of *Wolbachia* was higher in the infected group (*Moussa et al., 2020*). Meanwhile, in *Diaphorina citri* (Psyllidae family), *Wolbachia* has a higher proportion in the presence of 'Ca. Liberibacter asiaticus' than uninfected psyllids (*Song et al., 2019*; *Jiang et al., 2023*). Given similar results, some authors suggest a potential interaction between *Wolbachia* and members of the microbiome of *D. citri* (*Fagen et al., 2012*; *Hosseinzadeh et al., 2019*). Intriguingly, in both studies and in the results observed in this work, plant pathogenic bacteria affected *Wolbachia*. In insect vectors, *Wolbachia* may influence aspects of host biology. In *B. cockerelli*, the *Wolbachia*-infected central and western haplotypes were more likely to harbor and transmit 'Ca. Liberibacter' compared with the *Wolbachia*-noninfected northwestern haplotype (*Cooper et al., 2023*). This last piece of evidence points to the possibility *Wolbachia* may be a key element in the microbiome in the context of pathogen transmission processes by insect vectors. We still know little about how symbionts interact with one another within a host and whether the particularly intimate interaction between members of the host's microbiome has an effect on symbiont density. Future studies are needed with controlled infection of the group of *B. cockerelli* and techniques specifically quantitative ones, for a concrete analysis of what was observed in the present study based on OTU relative abundance.

## CONCLUSIONS

Our data support that the members with the higher representation in the microbial community of *B. cockerelli* central haplotype, comprise their obligate obligate-symbionts,

*Carsonella*, and facultative symbionts *Wolbachia*. The endosymbiont bacteria *Wolbachia* strain B was detected through PCR in psyllid identified as *B. cockerelli* Central haplotype in samples collected from four locations in Mexico. Support the evidence that have been seen in previous studies in psyllids. We also found that with the NGS technology was possible to identify OTU of the plant pathogen '*Ca*. Liberibacter'. Likewise, the presence of the '*Ca*. Liberibacter' plant pathogen affects the structure of the bacterial community associated with the *B. cockerelli* Central haplotype.

## ACKNOWLEDGEMENTS

We thank Paul Gaytan, Jorge Yañéz, and Eugenio López for primer synthesis, and sequencing at Instituto de Biotecnología, Universidad Nacional Autónoma de México. Bruno Gomez-Gil for 16S V3 rDNA sequencing at Laboratorio de Genómica Microbiana, CIAD-Mazatlán, México. We also thank to Angel Edgardo Carrillo García and Gil Ezequiel Ceseña Beltran, CIBNOR, for the assistance.

### Funding

Current investigations from the group are supported by CONAHCYT/México. There was no additional external funding received for this study. The funders had no role in study design, data collection and analysis, decision to publish, or preparation of the manuscript.

### Grant Disclosures

The following grant information was disclosed by the authors:
CONACYT/México.

### Competing Interests

The authors declare that they have no competing interests.

### Author Contributions

- Maria Goretty Caamal-Chan conceived and designed the experiments, performed the experiments, analyzed the data, prepared figures and/or tables, authored or reviewed drafts of the article, and approved the final draft.
- Aarón Barraza conceived and designed the experiments, performed the experiments, analyzed the data, prepared figures and/or tables, authored or reviewed drafts of the article, and approved the final draft.
- Abraham Loera-Muro analyzed the data, authored or reviewed drafts of the article, and approved the final draft.
- Juan J. Montes-Sánchez analyzed the data, authored or reviewed drafts of the article, and approved the final draft.
- Thelma Castellanos analyzed the data, authored or reviewed drafts of the article, and approved the final draft.
- Yolanda Rodríguez-Pagaza analyzed the data, authored or reviewed drafts of the article, and approved the final draft.

## Data Availability

The data is available at the National Center for Biotechnology information: PRJNA889072.

The Wolbachia endosymbiont *Bactericera cockerelli* outer membrane protein (*wsp*) gene data are available at GenBank/EMBL/DDBJ: OP071082.1, OP071083.1, OP071084.1 and OP071085.1.

The Bactericella cockerelli mtCOI gene data is also available at GenBank: OP019751.1, OP019752.1, OP019753.1 and OP019754.1.

## Supplemental Information

Supplemental information for this article can be found online at http://dx.doi.org/10.7717/peerj.16347#supplemental-information.

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
