# Peer review of "Bacterial communities of the psyllid pest Bactericera cockerelli (Hemiptera: Triozidae) Central haplotype of tomato crops cultivated at different locations of Mexico"

_PeerJ, doi:10.7717/peerj.16347_

## Round 0.1 · original submission · Major Revisions

This is an interesting manuscript that deals with the microbiome of the psyllid (please, correct the typo “pshyllid” in the title) Bactericera cockerelli, a pest of different crops and vector of the plant pathogen Candidatus Liberibacter, in 4 different populations from Northern and Central Mexico. Intriguingly, the authors found that the presence of the plant pathogen seems to affect the insect’s bacterial community. However, the manuscript needs some improvement.

For example, the introduction should be better organized and should contain the most relevant literature on psyllid microbial communities.

Materials and methods need more details, as suggested by reviewer 2 (regarding the species identification by using body colors, I agree with reviewer 2 that in insects they are not suitable for diagnosis, especially in mounted specimens; however, in B. cockerelli the body markings are quite stable and generally used to distinguish the species from other psyllids. By the way, did the authors checked also other morphological characters e.g., apical spurs on the tibiae, size and morphology of wing cells, etc.?).

The discussion should be rewritten, as it appears a little repetitive and not always focused on the interpretation of the results. In my opinion, there is no need for further analysis (e.g., qPCR diagnosis to corroborate the OTU relative abundance results), but the authors should consider in the discussion the potential limits of their analysis.

The authors should also discuss the fact that the Neorickettsia genus was dominant. Regarding Wolbachia, I suggest the author to include in the discussion previous recent literature demonstrating a pattern between infection with the plant pathogen and Wolbachia (doi: 10.1094/PDIS-11-22-2701-RE).

More details are also needed regarding the occurrence of the different haplotypes (to be considered in the discussion).

In the conclusion, I suggest the authors be more cautious when they state that their data “support the vertical transmission”: this was not the aim of their work nor they provided direct evidence of this mode of acquisition/transmission of the bacterial community.

Finally, I suggest the authors carefully edit the numerous typos (and the structure of many sentences) present in the manuscript before resubmission.

Reviewer 1 ·

Basic reporting

This paper describes the microbiota composition of different psyllid populations collected in Mexico, examining the presence of differences according to several parameters (geographical position, genetic background, Wolbachia infection, Liberibacter infection). The cited literature is generally appropriate and complete, and figures are correctly prepared. However, in the introduction I have remark: In the first part of the introduction (L60-80), it is not clear when the authors are treating the microbiota of arthropods/insects in general, when they refer to sap-feeding insects (Hemiptera) in detail, and when they refer to vectors specifically. I suggest organizing the initial part of the text describing bacterial associates of insects in general, then explain the features of endosymbionts of hemipterans, and finally explain how they affect the vectors specifically.
The English language should be improved to ensure that an international audience can clearly understand your text.

Experimental design

Overall, the experimental design is correctly presented. My only remark regards L117: please specify what was the collection date/period

Validity of the findings

One of the most relevant results of this work is the separation between Liberibacter high carrier and low carrier insects, in front of a generally reduced variability of all the other tested parameters (i.e., insect diversity, Wolbachia diversity). However, it appears that the authors did not exhaustively treat this result. First, the OTU relative abundance is not the most solid technique for quantifying a single component of the microbiota. Moreover, since a major pest trait of B. cockerelli is its vector status of Liberibacter, this association should be investigated with more details to improve the quality of this work. I suggest to perform a qPCR diagnosis to corroborate the OTU relative abundance results. Additionally, the Liberibacter infection in tested samples should be further investigated: did you find one or both haplotypes that are known to be related to this psyllid (LsoA and B)? If 2 haplotypes were present, did you find any difference in the OTU composition according to the Liberibacter haplotype? Since these two haplotypes were shown to induce different effects on the vector’s fitness, we may expect some outcome also in the microbiota composition.

Reviewer 2 ·

Basic reporting

English should be improved in some parts of the text. More literature should be reviewed and included into the text, particularly the discussion part.

Experimental design

Please add more information on the analyses and statistics you used into the Material and Methods section. Some analyses are mentioned in the Results section for the first time.

Validity of the findings

Discussion and a part of conclusion should be rewritten. Recommendations and suggestions are available are available in the additional comments.

Additional comments

This is an interesting manuscript on the microbial communities of vector psyllid Bactericera cockerelli. However, I think that at the moment the current version of the manuscript is lacking a significant amount of literature on psyllid microbial communities and discussion should be rewritten to address your important findings in a more clear way.

52. In the sentence “‘Candidatus Liberibacter’ influences the microbiota structure of insects” not of insects but of this particular psyllid species.

76. Change insects to phytophagous insects. Also, please provide the references to the statements in lines 76-77.

81. Triozidae should not be in italics. In contrast to bacterial families, animal and plant families are used without italics. Please check across the whole text.

92-94. Please reformulate the sentence.

94-97. References are missing.

98. By vertical not vertically.

101. In the part “northwestern and southwestern haplotypes are Wolbachia free”, I would not refer to these haplotypes as “free” but there was rather no Wolbachia found. The word “free” is a bit confusing.

104-107. Please reformulate the sentences. English is not clear.

115-116. Did you use any other characters and literature for the species identification. The colour description is not a good diagnostic character since it changes in response to the mounting medium, etc.

145-146. To reconstruct the phylogenetic analyses, it is always better to employ proper analyses and tools such as IQTREE for Maximum likelihood and Mr Bayes for Bayesian Inference.

162. Reference for the used primer is missing.

212-213. Please explain in details based on what ground so many reads were eliminated.

215-216. On second reference across the whole text, ‘Candidatus Liberibacter’ can be shortened to ‘Ca. Liberibacter’ or even to Liberibacter

225. Independent on the location/locality.

257. A general finding of Neorickettsia in such high relative abundance is interesting. Please elaborate this finding in the Discussion part.

266. Please provide a full name, also Solanaceae should be without italics. Provide the references.

289. 16S rRNA

291-292. Please check the grammar.

General comment to the Discussion: in the discussion we aim to interpret the results (i.e. it is not necessary to write again all names of the performed analyses, etc.). In your discussion, you mostly repeat the results section. In my opinion, the discussion part should be mostly rewritten. You should rather try to interpret your findings by comparing your results with literature available.


295-296. Please expand on this. There are also many other studies, which provided the metabarcoding data on microbial communities in psyllids. More literature needs to be studied.

312. psyllids

329-330. Wolbachia is not usually transmitted vertically but rather horizontally. Please explain your idea, expand and provide references.

357-358. This is too strong to talk about vertical transmission of bacteria found in your study. You have not performed any acquisition or transmission experiment, neither you had a very high number of psyllid replicates to suggest any vertical transmission. However, it was not the main aim of your study. Please elaborate your conclusion based on your real findings and results.

General comment to figures. Please use a psyllid name in full. When you list the name of a single species as an outgroup, please use a singular form of the verb to be (was instead of were).

526. The name of the phylogenetic analysis should be mentioned.

528 and 533. Bactericella maliculipennis should be fully in italics.

543-544, 561. Candidatus should be in italics.

Figures 2 and 3. What is highlighted in green and pink, respectively? Please add. Also, please add the name of the phylogenetic analysis.

Figure 3. Also, if you make the supergroup identification of your Wolbachia strain, you should mark in the tree other supergroups to make it easier to follow your identification. In addition, it would be also nice to include in your analyses the representatives of other supergroups than just solely A and B.

---

## Round 0.2 · accepted · Accept

The authors have addressed the reviewers' comments and in my opinion, the manuscript is ready to be published.

Reviewer 1 ·

Basic reporting

The authors provided satisfactory responses to my previous comments, so I believe that the manuscript has now been improved and is suitable for publication.

Experimental design

-

Validity of the findings

-